# Aquatic Therapy versus Standard Rehabilitation after Surgical Rotator Cuff Repair: A Randomized Prospective Study

**DOI:** 10.3390/biology11040610

**Published:** 2022-04-17

**Authors:** Adrien Dufournet, Xue Ling Chong, Adrien Schwitzguébel, Corinne Bernimoulin, Myriam Carvalho, Hugo Bothorel, Alexandre Lädermann

**Affiliations:** 1Physiotherapy Department, Hirslanden La Colline Clinic, 1206 Geneva, Switzerland; Adrien.Dufournet@hirslanden.ch; 2Division of Orthopaedics and Trauma Surgery, La Tour Hospital, Rue J.-D. Maillard 3, 1217 Meyrin, Switzerland; chongxueling@gmail.com; 3AS Sports Medicine, Faubourg de l’Hôpital 81, 2000 Neuchâtel, Switzerland; adrien.schwitzguebel@gmail.ch; 4Physiotherapy Department, Geneva University Hospitals, 1205 Geneva, Switzerland; corinnebernimoulin@gmail.com; 5Physiotherapy Department, La Tour Hospital, 1217 Meyrin, Switzerland; carvalho.myriam@gmail.com; 6Research Department, La Tour Hospital, 1217 Meyrin, Switzerland; hugo.bothorel@latour.ch; 7Division of Orthopaedics and Trauma Surgery, Geneva University Hospitals, 1205 Geneva, Switzerland; 8Faculty of Medicine, University of Geneva, 1205 Geneva, Switzerland

**Keywords:** shoulder, aquatic therapy, rehabilitation, outcomes, PROMs, hydrotherapy, RCR, rotator cuff tear repair, tendon healing

## Abstract

**Simple Summary:**

Rotator cuff lesion is a common shoulder condition that can cause significant pain and functional impairment. Arthroscopic rotator cuff repair is often performed when conservative treatment has failed, but there is no consensus established for optimal postoperative rehabilitation. In this prospective randomized control study (86 patients), we compared the standard land-based rehabilitation to aquatic therapy but found no significant differences in terms of range of motion, function, and pain at 1.5, 3, 6, and 24 postoperative months. Likewise, both groups were comparable in terms of postoperative tendon healing, complications, workstop duration, and patient satisfaction. Thus, the systematic use of immediate aquatic therapy after surgery does not seem necessary, though further studies can be performed to specifically identify the type of patients who could greatly benefit from it.

**Abstract:**

Introduction: Post-operative rehabilitation following rotator cuff tear repair (RCR) is important to promote tendon healing, restore strength, and recover normal function. Aquatic therapy in hot water allows body relaxation, which promotes patient conditioning for efficient rehabilitation. The aim of this study was to assess whether aquatic therapy is more efficient than standard (land-based) rehabilitation in terms of range of motion (ROM), function, and pain after arthroscopic RCR. Methods: We prospectively randomized 86 patients scheduled for arthroscopic RCR to either aquatic therapy (*n* = 44) or standard rehabilitation (*n* = 42) using block sizes of four or six. Patients were evaluated clinically at 1.5, 3, 6, and 24 months and using ultrasound (US) at 6 months. Two-way mixed ANOVA tests were performed to evaluate the effects of rehabilitation type (between-subjects factor) on ROM and patient reported outcome measures (PROMs) over time (within-subjects factor). Post-hoc inter-group comparisons at each time point were also conducted using Wilcoxon rank sum tests or unpaired Student *t*-tests and adjusted for multiple comparisons using the Bonferroni correction. Results: The two groups did not differ significantly in terms of demographic data or pre-operative characteristics, except for the Single Assessment Numeric Evaluation (SANE) score, which was lower in the aquatic therapy group (37.9 ± 23.6 vs. 55.6 ± 24.9, *p* = 0.019). The mixed model revealed the absence of interaction effect between the type of rehabilitation and time on PROMs and ROM except on the SANE score (*p* < 0.001), which was biased by the existing pre-operative difference mentioned above. Furthermore, none of the post-operative outcomes were statistically different between the two groups at 1.5, 3, 6, and 24 months. In addition, no significant difference could be noted regarding tendon healing rate (*p* = 0.443), complication (*p* = 0.349), workstop duration (0.585), or patient satisfaction (*p* = 0.663). Conclusion: Compared to the standard rehabilitation, the aquatic therapy did not yield superior clinical and functional outcomes after arthroscopic RCR when started immediately after the surgery.

## 1. Introduction

The lesion of the rotator cuff is a frequently encountered condition that increases with age [1] and can cause significant pain and functional impairment [2]. Arthroscopic rotator cuff repair (RCR) is a common procedure used when conservative management has failed. Optimizing the outcome after RCR is a challenge for patient, surgeon, and physiotherapist. Post-operative rehabilitation following RCR is often recommended to promote tendon healing, restore strength, and recover normal function. Although there are many techniques for RCR developed over the last 20 years, there are very few studies published on post-operative rehabilitation [3].

The goal of rehabilitation is to restore a functional shoulder in sufficient time [4]. The key is to focus on the mobilization of the shoulder complex while avoiding excessive stress on the repaired tendon [5]. Aquatic therapy may offer an advantageous alternative to the conventional land-based therapy. The former, also called hydrotherapy, involves performing exercises in the water rather than on land. It is differentiated from balneotherapy in that there are no specific mineral additives in the water. It facilitates buoyancy to reduce body weight, neutralizing forces acting on the joint, which can optimize physiologic muscle activation patterns [6]. Hot water also allows body relaxation and well-being that promote patient conditioning for effective rehabilitation. Water-based exercises can allow active motion to begin at an earlier stage than land-based exercises [6,7] without compromising repair integrity [8,9].

Aquatic therapy has already been established for rehabilitation after knee and hip surgeries [10,11], or for patients with musculoskeletal conditions [12]. There is increasing interest in aquatic therapy for post-op rehabilitation after surgery for rotator cuff lesions. However, there is insufficient evidence regarding the optimal rehabilitation modality following RCR [9]. Moreover, existing comparative studies on this topic have been based on a small number of patients [6,7] using aquatic therapy in combination with standard (land-based) therapy as the experimental group [7] or differing in terms of immobilization duration following surgery [13]. Therefore, new high methodological quality randomized trials are needed to establish a possible benefit of the therapy on pain, mobility, shoulder function, and quality of life of the patient.

Hence, the purpose of this study was to compare the clinical and radiological outcomes of arthroscopic RCR over a period of two years between two different post-operative rehabilitation modalities: aquatic therapy and land-based therapy. The hypothesis is that aquatic therapy would provide faster recovery than standard rehabilitation.

## 2. Materials and Methods

Between May 2018 and November 2019, all patients who had a primary RCR performed by the senior author (A.L.) were considered potentially eligible for inclusion in this prospective study. Inclusion criteria were small to large sized symptomatic supraspinatus and/or infraspinatus lateral tendon disruption (B1) [14], grade 1 to 2 tendon retraction according to Patte [15], and fatty infiltration stage ≤2 according to Goutallier [16]. We excluded (1) patients unable to follow the study protocol, (2) other types of rotator cuff lesion (bony rotator cuff (A), medial tendinous disruption (B2), tendon-to-tendon adhesion ‘Fosbury flop tear’ (B3), and musculotendinous junction lesion (C type)) [14], (3) patients with subscapularis tendon lesion, (4) associated superior labrum anterior posterior (SLAP) lesion, or (5) frozen shoulder [17]. All patients were asked to fill an informed consent form for their participation in this study, which had been approved by our institutional review board (CCER 2016-02242), and registered at our National Clinical Trials Portal (SNCTP No. 000002244) and at ClinicalTrials.gov (NCT05106842).

### 2.1. Pre- and Post-Operative Clinical Assessment

Data collection was conducted by an independent observer (C.B.) before surgery and at 1.5, 3, 6, and 24 months post-operatively. Patient characteristics included age at index surgery, sex, body mass index (BMI), and operated side (dominant or not). The outcomes of interest were the range of motion (ROM), which includes active forward flexion (AFF) and external rotation (ER), as well as the patient-reported outcome measures (PROMs) that are commonly used to evaluate the evolution of patients undergoing arthroscopic RCR: the Single Assessment Numeric Evaluation (SANE) score (worst 0–100 best) [18], the Constant score (worst 0–100 best) [19], the pain assessed on a visual analog scale (pain on VAS, best 0–100 worst), the American Shoulder and Elbow Surgeons (ASES, worst 0–100 best) [20], and the simple shoulder test score (SST, worst 0–12 best) [21]. It is worth noting that the ASES and STT scores were only evaluated 2 years after surgery, and that tendon healing was assessed at 6 post-operative months using ultrasound (US) [22] and categorized using the Sugaya classification [23]. The workstop duration was also collected for non-retired patients.

### 2.2. Randomization Process

To allocate patients into two groups, the investigators generated a random list of numbers with an allocation of 1:1 using block sizes of four or six. No stratification techniques were used in the randomization process.

### 2.3. Surgical Procedure

The surgery was performed under general anesthesia and U.S.-guided interscalene brachial plexus block with the patients placed in a beach-chair position. Adjuvant acromioplasty was performed only in patients who had radiographic signs of dynamic impingement [24], and resection of the distal part of the clavicle was performed when pain was elicited by palpation of the acromioclavicular joint. Biceps tenodesis or tenotomy was performed when the posterior wall of the bicipital groove was damaged. All repairs were carried out using two anchors, of which one was implanted at the bone–cartilage junction, and one was implanted at the lateral part of the greater tuberosity [25]. At the end of the intervention, all repairs were complete and “watertight,” with adequate restoration of the tendons to their footprints. Post-operative care included regular wound dressing twice per week with removal of skin closure sutures 10 days after surgery.

### 2.4. Rehabilitation Protocol

Each patient was required to wear a universal sling for four weeks to keep the shoulder immobilized in an internally rotated position. During their hospital stay, all patients received identical surgeon recommendations to perform self-mobilization of the shoulder five times a day. After skin closure removal at 10 days, formal physiotherapy was initiated. Aquatic therapy was performed in a swimming pool (depth 125–140 cm, temperature 34 degrees Celsius under supervision by a physiotherapist (A.D. and M.C.). The patients were asked to kneel or sit to submerge both shoulders and perform exercises consisting of progressive passive motion of the shoulder for three weeks (Figure 1) followed by active assisted rehabilitation. Land-based therapy was performed at a rehabilitation center, also with supervision by a physiotherapist. The patients were taught a similar protocol consisting of progressive passive motion for three weeks followed by active-assisted motion of the shoulder. Strengthening exercises began at three months in both groups [26].

### 2.5. Statistical Analysis

Based on a pilot study, the post-operative patient AFF following standard rehabilitation after RCR was 105 ± 28° at 6 weeks. For the present study, a sample size calculation was performed *a priori* to ensure the detection of 20° of difference in AFF between groups, considered as the minimal clinically important difference (MCID). With a statistical power of 0.80, a significant level of 0.05 and an anticipated drop-out rate of 25%, the sample size needed was estimated at 86 patients. 

The missing information (representing less than 2% of the data) was completed using either the last observation carried forward when possible or by a multiple imputation by chained equation (MICE) method. Descriptive statistics were used to summarize the data. Continuous variables were reported as mean ± standard deviation (min-max) and categorical data were reported as proportions. The normality of continuous variable distributions was assessed by the Shapiro–Wilk test and the normality of the residuals was visually assessed on a Q–Q plot. Two-way mixed ANOVA tests were performed to evaluate the effects of rehabilitation type (between-subjects factor) on ROM and patient reported outcome measures (PROMs) over time (within-subjects factor). Post-hoc inter-group comparisons at each time point were also conducted using Wilcoxon rank sum tests or unpaired Student t-tests and adjusted for multiple comparisons using the Bonferroni correction. The analyses were performed using R (version 3.6.2, R Foundation for Statistical Computing, Vienna, Austria) and following the intention to treat analysis method, known to avoid any bias in superiority trials; *p*-values of <0.05 were considered significant.

## 3. Results

A total of 92 patients met the eligibility criteria, but 6 (6.5%) were excluded because they declined to participate in the study (Figure 2). This left a final cohort of 86 patients who underwent aquatic therapy (*n* = 44) or land-based therapy (*n* = 42) without any significant difference in terms of age (55.0 ± 10.4 vs. 58.5 ± 9.8, *p* = 0.107), BMI (25.4 ± 4.1 vs. 27.4 ± 16.9, *p* = 0.739), sex (61.9% vs. 59.5% men, *p* = 1.000), and affected side (76.2% vs. 69.0% dominant, *p* = 0.813) (Table 1).

### Clinical Outcomes

Pre-operatively, the two groups were comparable in terms of ROM and clinical scores, except for the SANE score, which was statistically lower in the land-based group (37.9 ± 23.6 vs. 55.6 ± 24.9 vs, *p* = 0.019) (Table 2).

The two-way mixed models revealed the absence of interaction effects between the type of rehabilitation and time on PROMs and ROM, except for the SANE score (F = 6.904, *p* < 0.001), explained by the pre-operative SANE score difference between groups cited above. Post-hoc tests confirmed that the two groups did not statistically differ from each other in PROMs (Figure 3) and ROM (Figure 4) at the different follow-up time points.

At two post-operative years, the two groups had comparable ASES (*p* = 0.864) and SST scores (0.846). There were also no statistical differences in workstop duration (*p* = 0.580), satisfaction scores (*p* = 0.663), or tendon healing rate (*p* = 0.443) (Table 3). Complications were noted for three patients in the standard group vs. one in the aquatic therapy group (7.3% vs. 2.8%, *p* = 0.349). The complications in the aquatic therapy group included a frozen shoulder and those in the standard group were painful acromioclavicular joint, iterative rotator cuff retear, and a revision to reverse total shoulder arthroplasty.

## 4. Discussion

There is increasing interest in aquatic therapy for post-operative rehabilitation for surgeries such as RCR in view of positive results from studies on knee and hip arthroplasties [10,11]. With its distinctive advantage in avoiding stress on rotator cuff tendon repair, aquatic therapy was a good modality to be explored in post-operative rehabilitation. However, the principal finding of our study was that aquatic-based therapy, if started soon after surgery, was not associated with better functional outcomes compared to standard (land-based) therapy, thereby invalidating our hypothesis.

A review of existing literature did not yield any other studies that specifically address the comparison of the therapies after rotator cuff repair [27]. Brady et al. has shown that combined aquatic therapy and land-based therapy can improve passive flexion ROM measures at three and six weeks [7]. However, this study was performed on a small number of patients (*n* = 18) and compared land as well as combined aquatic and land post-operative rehabilitative therapy. A prospective case study carried out on five patients with early addition of a comprehensive aquatic-assisted exercise program at two weeks to land-based therapy after surgery has also shown promising improvement in Shoulder Pain and Disability Index and Penn Shoulder Score as well as shoulder strength and ROM [6]. Therefore, future studies could explore the use of aquatic therapy to initiate active ROM exercises before four post-operative weeks to reduce shoulder stiffness without compromising repair integrity.

The availability of swimming pools designated for aquatic rehabilitation is parsimonious. Our prospective study showed that aquatic therapy does not have a significantly better clinical, functional, or radiological outcome, compared to land-based therapy. This finding is interesting as it confirms that traditional land-based therapy remains the gold standard. However, in the present study, rehabilitation was initiated soon after the surgery. In another prospective and comparative study by Lädermann et al. [13], aquatic therapy initiated one month post-operatively had higher Constant and SANE scores compared to those performing land-based therapy or self-rehabilitation therapy. Consequently, the benefit of aquatic therapy in the present study could have been occulted by a treatment initiated precociously. Moreover, aquatic therapy, with the benefits of buoyancy to buffer the effects of forces and yet not compromise on RCR integrity, seems a good modality to use for post-operative rehabilitation if the facilities are available. Therefore, aquatic therapy could be useful for patient groups at risk for post-operative stiffness, such as small and partial articular-sided (PASTA) tears, workers’ compensation, age less than 50 years, or concomitant labral repair [28,29].

### Limitations

Our limitations are that both patients and physiotherapists administering intervention were not blinded to the therapy due to the different set up in each intervention. We attempted to reduce bias elsewhere by introducing an independent observer (C.B.) who was blinded to the treatment administered when collecting data. Moreover, further studies with a greater cohort size are needed to analyze the differences in terms of complications, satisfaction, or tendon healing. The strengths of the present study are its prospective randomized design, based on a consecutive series of patients operated on by the same senior surgeon (A.L.) within a short inclusion period.

## 5. Conclusions

When initiated immediately after surgery, aquatic therapy did not yield superior clinical or functional outcomes compared to land-based rehabilitation after arthroscopic RCR. Further studies are required to better define when the aquatic therapy should begin and on which patients in order to optimize both patient outcomes and unnecessary costs.

## Figures and Tables

**Figure 1 biology-11-00610-f001:**
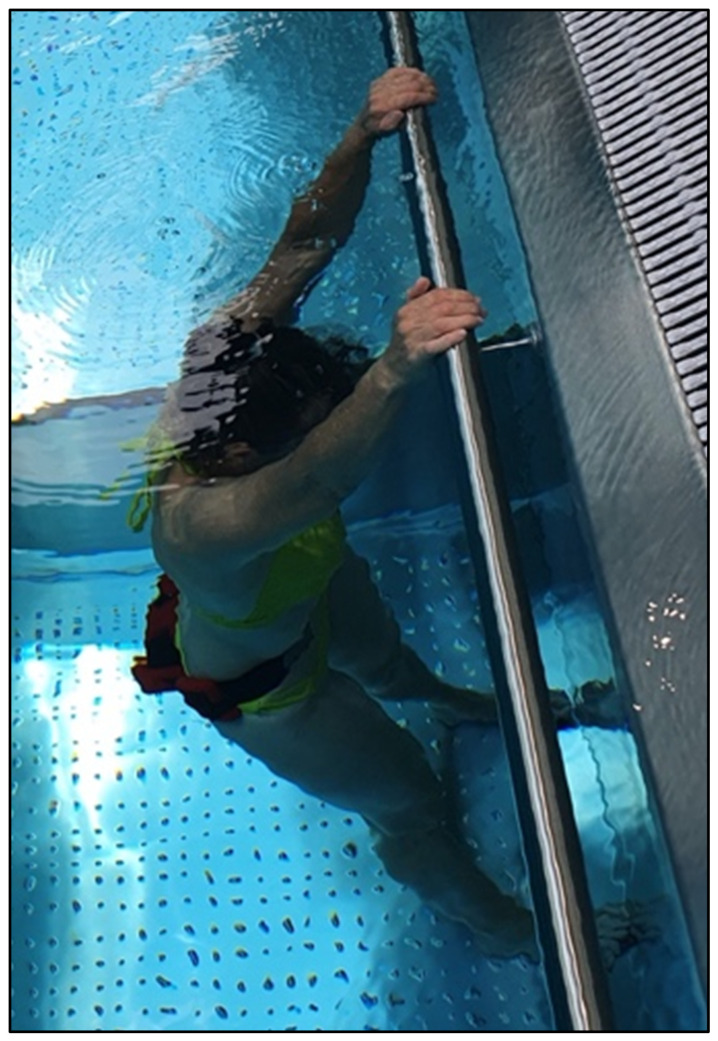
Passive motion of the shoulder (aquatic therapy).

**Figure 2 biology-11-00610-f002:**
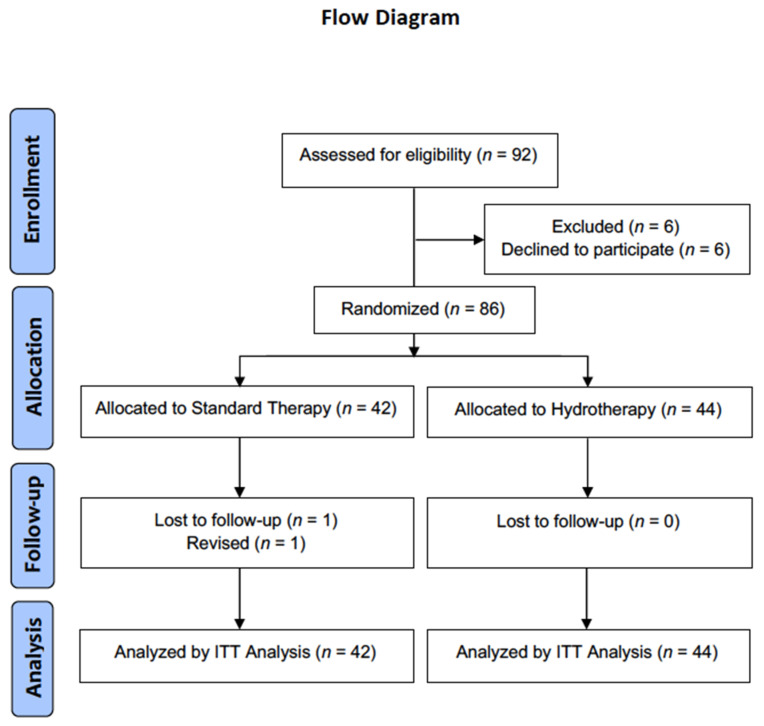
Consort flow diagram.

**Figure 3 biology-11-00610-f003:**
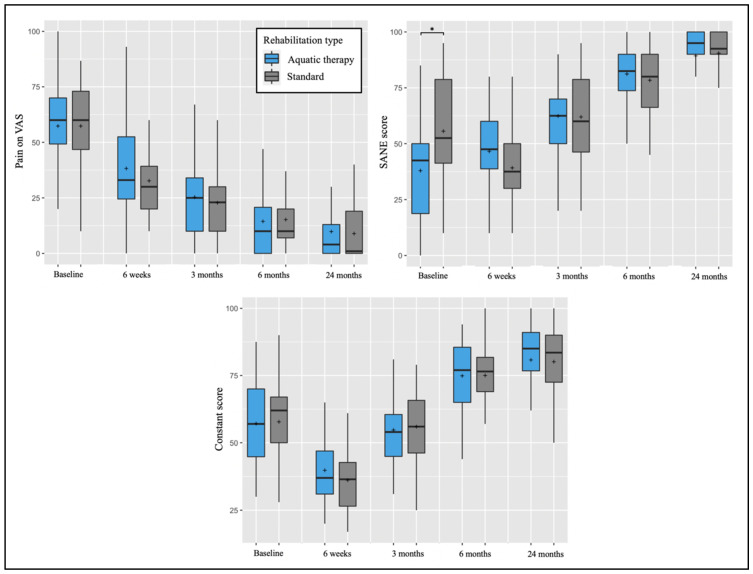
Pre-operative and post-operative pain on VAS, SANE, and Constant scores depending on post-operative rehabilitation (aquatic therapy vs. standard). The plots illustrate median values (bold lines), means (+), interquartile ranges (boxes), and 95% CIs (whiskers). * A significant difference was found between groups at the same follow-up point.

**Figure 4 biology-11-00610-f004:**
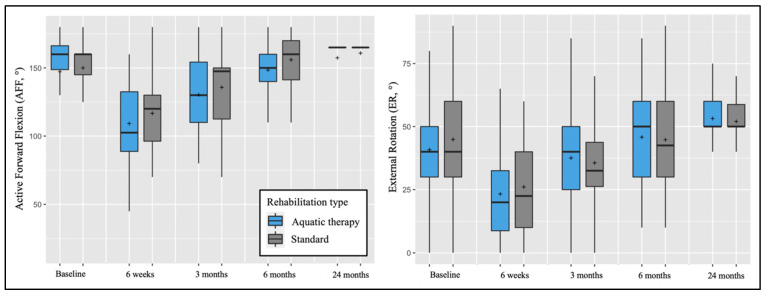
Pre-operative and post-operative active forward flexion (AFF) and external rotation (ER) depending on post-operative rehabilitation (Aquatic therapy vs. standard). The plots illustrate median values (bold lines), means (+), interquartile ranges (boxes), and 95% CIs (whiskers).

**Table 1 biology-11-00610-t001:** Demographic data.

	Standard Therapy (*n* = 42)	Aquatic Therapy (*n* = 44)	*p*-Value
Age	58.5 ± 9.8	55.0 ± 10.4	0.107
BMI	27.4 ± 16.9	25.4 ± 4.1	0.739
Male sex (%)	59.5	61.9	1.000
Dominant side (%)	69.0	76.2	0.813

**Table 2 biology-11-00610-t002:** Comparisons of pre- and post-operative clinical outcomes between the standard or aquatic therapy groups.

	Standard (*n* = 42 Patients)		Aquatic Therapy (*n* = 44 Patients)		*p*-Value *
	Mean ± SD	(Range)	Mean ± SD	(Range)	
**AFF (°)**					
Baseline	150.0 ± 28.2	(15.0–180.0)	147.2 ± 32.7	(35.0–180.0)	1.000
6 weeks	116.7 ± 26.4	(70.0–180.0)	109.2 ± 30.8	(45.0–160.0)	1.000
3 months	135.8 ± 30.5	(70.0–180.0)	130.3 ± 28.1	(80.0–180.0)	1.000
6 months	156.0 ± 19.5	(110.0–180.0)	148.4 ± 18.9	(100.0–180.0)	0.225
2 years	160.8 ± 15.9	(75.0–180.0)	157.4 ± 22.3	(45.0–165.0)	1.000
**ER (°)**					
Baseline	44.9 ± 22.3	(0.0–90.0)	40.8 ± 15.8	(0.0–80.0)	1.000
6 weeks	26.1 ± 19.8	(0.0–90.0)	23.3 ± 20.7	(0.0–70.0)	1.000
3 months	35.6 ± 18.4	(0.0–90.0)	37.5 ± 19.7	(0.0–90.0)	1.000
6 months	44.8 ± 18.0	(10.0–90.0)	45.8 ± 19.0	(10.0–85.0)	1.000
2 years	52.0 ± 13.6	(0.0–90.0)	53.2 ± 13.4	(10.0–85.0)	1.000
**Pain on VAS**					
Baseline	57.3 ± 18.3	(10.0–86.7)	57.3 ± 20.0	(6.0–100.0)	1.000
6 weeks	32.7 ± 18.2	(10.0–90.0)	38.3 ± 21.3	(0.0–97.0)	0.928
3 months	22.8 ± 14.3	(0.0–60.0)	25.3 ± 16.3	(0.0–67.0)	1.000
6 months	15.2 ± 14.9	(0.0–60.0)	14.4 ± 15.1	(0.0–53.0)	1.000
2 years	8.9 ± 11.7	(0.0–40.0)	9.8 ± 14.0	(0.0–60.0)	1.000
**SANE score**					
Baseline	55.6 ± 24.9	(10.0–95.0)	37.9 ± 23.6	(0.0–85.0)	0.019
6 weeks	39.2 ± 16.7	(10.0–80.0)	46.6 ± 18.0	(5.0–80.0)	0.306
3 months	62.0 ± 19.2	(20.0–95.0)	62.4 ± 14.4	(20.0–90.0)	1.000
6 months	78.4 ± 15.3	(45.0–100.0)	81.2 ± 14.0	(50.0–100.0)	1.000
2 years	90.5 ± 15.8	(10.0–100.0)	89.3 ± 17.1	(20.0–100.0)	1.000
**Constant score**					
Baseline	57.8 ± 16.8	(21.0–90.0)	57.1 ± 16.0	(30.0–87.5)	1.000
6 weeks	36.2 ± 11.3	(17.0–61.0)	39.8 ± 13.6	(20.0–74.0)	1.000
3 months	56.0 ± 13.4	(25.0–79.0)	54.8 ± 12.5	(31.0–81.0)	1.000
6 months	75.0 ± 12.5	(40.0–100.0)	74.9 ± 12.9	(44.0–94.0)	1.000
2 years	80.1 ± 15.0	(18.0–100.0)	80.8 ± 16.3	(17.0–100.0)	1.000
**ASES Score at 2 years**	88.9 ± 16.0	(10.0–100.0)	88.4 ± 14.9	(33.0–100.0)	0.864
**SST score at 2 years**	10.0 ± 2.2	(3.0–12.0)	9.9 ± 2.4	(1.0–12.0)	0.846
**Workstop (weeks)**	11.9 ± 8.4	(0.0–28.0)	15.3 ± 13.1	(0.0–48.0)	0.585

* *p*-values obtained for repeated measurement comparisons were adjusted using the Bonferroni correction method. Underlined *p*-values indicate those below 0.05; ASES, American Shoulder and Elbow Surgeons, SST, simple shoulder test, SANE, Single Assessment Numeric Evaluation; VAS, visual analogic scale, AFF, active forward flexion; ER, external rotation

**Table 3 biology-11-00610-t003:** Comparisons of post-operative tendon healing and patient satisfaction between the standard and aquatic therapy groups.

	Standard (*n* = 42 Patients)		Aquatic Therapy (*n* = 44 Patients)		*p*-Value
	N	(%)	N	(%)	
**Sugaya classification**					0.443
Type 1	32	(76.2%)	39	(88.6%)	
Type 2	5	(11.9%)	4	(9.1%)	
Type 3	3	(7.1%)	1	(2.3%)	
Type 4	1	(2.4%)	0	(0.0%)	
Type 5	1	(2.4%)	0	(0.0%)	
**Satisfaction**					0.663
Very satisfied	32	(76.2%)	36	(81.8%)	
Satisfied	5	(11.9%)	6	(13.6%)	
Unsatisfied	4	(9.5%)	2	(4.5%)	

## Data Availability

Data supporting reported results can be requested to the corresponding author (alexandre.laedermann@gmail.com).

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
