# Peer review of "Aquatic Therapy versus Standard Rehabilitation after Surgical Rotator Cuff Repair: A Randomized Prospective Study"

_biology, 2022, doi:10.3390/biology11040610_

Round 1
Reviewer 1 Report
The authors propose a study to assess the efficiency of aquatic therapy under the rehabilitation of the rotator cuff tear repair (RCR). They propose a balanced distribution of patients between the control group (standard rehabilitation) and the test group (aquatic therapy). The patients were clinically evaluated over different months to evaluate the effects of rehabilitation type.
The abstract is very informative and concise. The authors provide sufficient background on the presented problem and the procedures carried out in the study are properly explained. They thoroughly presented that the initial hypothesis of the study was not supported by the discussed results. But, the conclusion of this article should be re-done. It is a two-line conclusion that has been previously stated. The authors should extend and emphasize the obtained results on their conclusions. For instance, by using the shown limitations at the end of the discussion and some of the information from the text above.
However, the following modifications are advised to be accomplished:
There are extra breaks along with the text. Is the journal’s template like this?
Please consider starting Section 2 with an introductory paragraph, instead of starting with “a title”.
Line 75 replace “The hypothesis was…” with “The hypothesis is…“.
Line 89 consider replacing “All patients gave written informed consent…” to “All patients were asked to fill an informed consent…”
Line 129: “Patients were protected for four weeks in a universal sling”? Please consider rephrasing this.
Paragraph on lines 167-173 needs to be reformulated. The authors showcase detailed and important characteristics of the studied patients that could be in a form of a table (easier to read) and it is what enables them to create a cohort.
Table 1 makes no sense to be over three pages. There are two many breaks. Must be changed.
Figures are not centered.
Table 2 needs to be changed as well.
Sentence on lines 212-213 “Aquatic-based therapy is a good avenue to explore to facilitate rehabilitation 212 with its protective benefits on avoiding stress on the tendon repair. ” needs to be rephrased.
In the paragraph formulated on lines 65-72, the authors could provide more insight if there is any work related to the one that they are proposing (e.g., has there been any previous study similar to this?). Later, the reader finds that they compare their study with others on the discussion. In my opinion, the paragraph on lines 228-239 doesn’t belong in discussion but on related work or the introduction. Or even, claims such as “Burmaster et al. [6] has suggested that hydrotherapy provided additional benefits of well-being.“ on lines 252-253 shouldn’t be here as well.
Author Response
|
Reviewer 1 |
|
|
|
Comments |
Answers |
Lines |
|
The abstract is very informative and concise. The authors provide sufficient background on the presented problem and the procedures carried out in the study are properly explained. They thoroughly presented that the initial hypothesis of the study was not supported by the discussed results. But, the conclusion of this article should be re-done. It is a two-line conclusion that has been previously stated. The authors should extend and emphasize the obtained results on their conclusions. For instance, by using the shown limitations at the end of the discussion and some of the information from the text above. |
Conclusion re-done. |
288-290 |
|
There are extra breaks along with the text. Is the journal’s template like this? |
Title deleted |
80 |
|
Line 75 replace “The hypothesis was…” with “The hypothesis is…“. |
Corrected. |
75-76 |
|
Line 89 consider replacing “All patients gave written informed consent…” to “All patients were asked to fill an informed consent…” |
Corrected |
92-93 |
|
Line 129: “Patients were protected for four weeks in a universal sling”? Please consider rephrasing this. |
Rephrased. |
132-133 |
|
Paragraph on lines 167-173 needs to be reformulated. The authors showcase detailed and important characteristics of the studied patients that could be in a form of a table (easier to read) and it is what enables them to create a cohort. |
Reformatted in a new table. (Table 1) |
179-181 |
|
Table 1 makes no sense to be over three pages. There are two many breaks. Must be changed. |
Table 1 reformatted to become Table 2. |
192 |
|
Figures are not centered. |
Figures centered. |
|
|
Table 2 needs to be changed as well. |
Table reformatted to Table 3. |
199-204 |
|
Sentence on lines 212-213 “Aquatic-based therapy is a good avenue to explore to facilitate rehabilitation 212 with its protective benefits on avoiding stress on the tendon repair. ” needs to be rephrased. |
Re-phrased. |
222-224 |
|
In the paragraph formulated on lines 65-72, the authors could provide more insight if there is any work related to the one that they are proposing (e.g., has there been any previous study similar to this?). Later, the reader finds that they compare their study with others on the discussion. |
Added |
Line 70-73 |
|
In my opinion, the paragraph on lines 228-239 doesn’t belong in discussion but on related work or the introduction. |
Deleted. It is better to focus the paper on aquatic vs land-based therapies and not on early mobilization.
|
Lines 228-38 |
|
|
Deleted since it is already in the introduction |
|
|
|
|
|
|
Reviewer 2 |
|
|
|
Please take a picture of figure 1 and replace it so that you can see it clearly.
|
Corrected |
|
|
There is absolutely no information about the participants. Please add basic information such as gender, no, height, and weight. Also, there should be no difference in basic information between the two groups. This is because, if there is a difference, it is likely that the results have been affected. |
Done. Created in Table 1. |
179-181 |
|
Check the references. |
Done. |
|
Reviewer 2 Report
Please take a picture of figure 1 and replace it so that you can see it clearly.
There is absolutely no information about the participants. Please add basic information such as gender, no, height, and weight. Also, there should be no difference in basic information between the two groups. This is because, if there is a difference, it is likely that the results have been affected. Check the references. Read the instruction to authors in detail.
Author Response

(The authors gave the same response as above.)

Round 2
Reviewer 2 Report
This study suggested very interesting results and experimental suggestions.
Also, this paper is well written with logical flow. Accept in present form.